# Continuous and Dynamic Circulation of West Nile Virus in Mosquito Populations in Bucharest Area, Romania, 2017–2023

**DOI:** 10.3390/microorganisms12102080

**Published:** 2024-10-17

**Authors:** Sorin Dinu, Ioana Georgeta Stancu, Ani Ioana Cotar, Cornelia Svetlana Ceianu, Georgiana Victorița Pintilie, Ioannis Karpathakis, Elena Fălcuță, Ortansa Csutak, Florian Liviu Prioteasa

**Affiliations:** 1Molecular Epidemiology for Communicable Diseases Laboratory, Cantacuzino National Military Medical Institute for Research and Development, 103 Splaiul Independenței, 050096 Bucharest, Romania; dinu.sorin@cantacuzino.ro; 2Department of Genetics, Faculty of Biology, University of Bucharest, 1-3 Aleea Portocalelor, 060101 Bucharest, Romania; stancu.ioanageorgeta@yahoo.com (I.G.S.); cs_ortansa@yahoo.fr (O.C.); 3Vector-Borne Infections Laboratory, Cantacuzino National Military Medical Institute for Research and Development, 103 Splaiul Independenței, 050096 Bucharest, Romania; cornelia_ceianu@yahoo.com (C.S.C.); georgiana.tiron09@gmail.com (G.V.P.); iannis.karpathakis@gmail.com (I.K.); 4Department of Microbiology, Faculty of Biology, University of Bucharest, 1-3 Aleea Portocalelor, 060101 Bucharest, Romania; 5Medical Entomology Laboratory, Cantacuzino National Military Medical Institute for Research and Development, 103 Splaiul Independenței, 050096 Bucharest, Romania; efalcuta@yahoo.com (E.F.); florianliviu@yahoo.com (F.L.P.)

**Keywords:** West Nile virus, lineage 2, *Culex pipiens*, Romania, continuous circulation

## Abstract

West Nile virus (WNV) is a mosquito-borne pathogen with a worldwide distribution. Climate change and human activities have driven the expansion of WNV into new territories in Europe during the last two decades. Romania is endemic for WNV circulation since at least 1996 when the presence of lineage 1 was documented during an unprecedented outbreak. Lineage 2 was first identified in this country during a second significant human outbreak in 2010. Its continuous circulation is marked by clade replacement, and even co-circulation of different strains of the same clade was observed until 2016. The present study aims to fill the information gap regarding the WNV strains that were circulating in Romania between 2017 and 2023, providing chiefly viral sequences obtained from mosquito samples collected in the Bucharest metropolitan area, complemented by human and bird viral sequences. WNV was detected mainly in *Culex pipiens* mosquitoes, the vectors of this virus in the region, but also in the invasive *Aedes albopictus* mosquito species. Lineage 2 WNV was identified in mosquito samples collected between 2017 and 2023, as well as in human sera from patients in southern and central Romania during the outbreaks of 2017 and 2018. Both 2a and 2b sub-lineages were identified, with evidence of multiple clusters and sub-clusters within sub-lineage 2a, highlighting the complex and dynamic circulation of WNV in Romania, as a consequence of distinct introduction events from neighboring countries followed by in situ evolution.

## 1. Introduction

West Nile virus (WNV) is the most widespread flavivirus globally [1], undergoing a rapid expansion in Europe in the last two decades, including in the northern latitude countries. This unprecedented geographic expansion, along with the recurrent detection of the same circulating strains, support the endemic presence of WNV on the continent [2,3]. The virus is transmitted in complex enzootic cycles between mosquitoes and birds and can cause lethal neuroinvasive disease in avian reservoirs, and in incidental hosts such as humans and equines. Mosquitoes from *Culex pipiens* (*Cx. pipiens*) species complex are the main vectors of WNV [4,5]. Up to nine lineages of WNV are recognized, and isolates belonging to lineages 1a and 2 are the main isolates responsible for human infections [3,5].

Before 1996, WNV was detected in Europe only in serosurveys studies in Albania (1958) [6] and France (1962) [7]. In 1996, an overwhelming human outbreak (352 cases of West Nile neuroinvasive disease—WNND and 17 fatalities) occurred in southeastern Romania [8]. Lineage 1a isolates highly similar to isolates circulating in Kenya, Senegal, Central African Republic, and Algeria were detected in *Cx. pipiens* vectors collected during this outbreak [9]. An even greater outbreak (480 human cases) with a higher case fatality rate occurred in 1999 in Russia and was caused by a highly similar strain [10]. The same year, a related WNV isolate was detected in a Chilean flamingo (*Phoenicopterus chilensis*) at the Bronx Zoo, New York City, representing the first record of this virus in North America [11].

Lineage 2 was detected in 2004 in Europe (Hungary), in a northern goshawk (*Accipiter gentilis*) [12] and then spread into the Balkans and caused a severe outbreak in 2010 in Greece [13]. The Nea Santa-Greece-2010 strain causing this outbreak [14] belonged to the central/southern European clade of lineage 2 [15], later named sub-lineage 2a [16]. Phylogenetic studies on lineage 2 WNV have shown that sub-lineage 2a isolates circulating in Europe are part of a monophyletic group resulting from a single introduction event that occurred in Hungary around 2002–2003 [16,17,18]. After its introduction to Europe, the sub-lineage 2a split into two clusters around 2006–2007, dispersing into northwest and western regions (cluster A) and southward (cluster B) [16,19]. Also, in 2010, a significant human outbreak (54 cases of WNND) was recorded in Romania and, unlike the 1996 epidemics, the cases were scattered across an extended territory of the country. In contrast to the Greece outbreak, a Volgograd 2007-like isolate belonging to the eastern European (Russian/Romanian) clade of lineage 2 [20], also known as sub-lineage 2b [16], was identified in a human sample during the Romania outbreak [21]. The continuous circulation of this strain in Romania was documented from 2011 to 2016, using human samples and mosquitoes collected in southeastern Romania, including Danube Delta [22,23]. Furthermore, the strain was identified in a *Hyalomma marginatum marginatum* tick collected from a song thrush (*Turdus philomelos*) near Danube Delta [20]. Afterwards, the co-circulation of the eastern European clade and the central/southern European clade strains was documented in Romania. The first isolates belonging to sub-lineage 2a were identified in Romania in human patients in 2015 and 2016, but during the same years they were also present in mosquito populations in Bucharest city and Danube Delta, co-circulating with sub-lineage 2b isolates [23]. In 2016, the endemically established Volgograd 2007-like strain was replaced by sub-lineage 2a strains. The event was associated with a severe human outbreak (93 cases of WNND) [23]. Following the introduction of lineage 2 in Europe, WNV spread rapidly and diverged, causing numerous outbreaks, culminating in the 2018 epidemics (1548 reported cases in EU/EEA). The following years were also marked by the intense circulation of the virus on the continent [2,3,24]. In addition to the intense circulation of WNV lineage 2 in Europe, lineage 1 was also reported in Italy [25] and Spain [26,27] in recent transmission seasons. The circulation dynamics of WNV in Romania followed the European trend, with significant number of cases during 2016–2019, 2022, and 2023 transmission seasons, with a peak in 2018 (267 WNND confirmed cases) [24,28,29,30,31,32,33,34].

Assessing intra- and inter-continental circulation of WNV by genomic surveillance addressing all the transmission cycle elements demonstrates the need for animal and public health interventions [3]. Complemented by vector control, monitoring of environmental factors [35], and efforts into vaccine development [36], this surveillance could prevent, or at least limit, epidemic events, especially in the context of climate-driven expansion of WNV [2].

We present here the molecular characterization of the mosquito WNV isolates circulating in the Bucharest city area over a seven-year period (2017–2023), and compare them with isolates from human and bird samples in Romania, collected in the same period.

## 2. Materials and Methods

### 2.1. Mosquito Collection, Identification, and Pooling

Mosquitoes were collected during seven years (2017–2023), between June and August/September/October in the Bucharest metropolitan area, which is situated in the Romanian plain, in the southeast of the country. The sampling sites are representative of a variety of urban and rural habitats for mosquito populations (Figure 1). The insects were collected using Centers for Disease Control and Prevention (CDC) gravid traps (John W. Hock Company, Gainesville, FL, USA). Adult mosquitoes were morphologically identified [37], and pooled in samples of maximum 60 specimens, according to species, sex, site and date of collection. Samples were kept at −70 °C until further processing.

### 2.2. Human Samples

Serum, urine, and brain tissue samples collected in 2017 and 2018 from patients with serologically confirmed WNV infections were included in this study. Human samples were not available for the rest of the study period. In total, there were 63 human specimens representing the 2017 outbreak and 28 human specimens from the 2018 outbreak tested.

### 2.3. Bird Samples

A serum sample obtained from a hooded crow (*Corvus cornix*) presenting signs of neurological disease, and a brain sample from a dead house sparrow (*Passer domesticus*) were collected in 2018 and analyzed in the study.

### 2.4. Real-Time PCR Detection of WNV

Viral RNA was extracted from mosquito pools using a QIAamp Viral RNA Mini kit (Qiagen, Hilden, Germany), as previously described [22], and used for detection of the WNV genome with a real-time RT-PCR kit (West Nile Virus Real-TM, Sacace Biotechnologies, Como, Italy) on a QuantStudio™ 5 Real-Time PCR System (Applied Biosystems, Waltham, MA, USA). Human and bird samples were processed using the same kits and equipment.

### 2.5. RAMP Assay for the Detection of WNV

The semiquantitative immunochromatographic screening test RAMP^®^ WNV kit (Response Biomedical Corporation, Burnaby, BC, Canada) was used following the instructions provided by the manufacturer, with the cut-off value and the grey zone interval previously recommended [38]. This assay uses fluorescent-dyed particles coated with anti-WNV antibodies, and is intended for testing mosquito pools and bird throat swabs.

### 2.6. Sanger Sequencing

Amplicons spanning a fragment of NS5 viral genomic region were amplified by PCR using primers FU2, cFD3 [39], and 9368f [12]. The latter two primers were used in a heminested PCR, only when it was necessary to enhance the sensitivity of the test. PCR products were sequenced using a BigDye™ Terminator v3.1 Cycle Sequencing Kit, on a SeqStudio Genetic Analyzer (Applied Biosystems, Waltham, MA, USA). Sanger sequences were visually inspected and edited with BioEdit version 7.7.1 [40].

### 2.7. Whole Genome Sequencing

WNV whole genome sequences were obtained using a metagenomic protocol. Briefly, the total RNA extracted from mosquito pools was used for double-stranded cDNA synthesis with Superscript II (Invitrogen, Applied Biosystems, Waltham, MA, USA) and a TruSeq RNA Library Prep Kit v2 (Illumina, San Diego, CA, USA). DNA libraries were obtained with a Nextera XT DNA Library Preparation Kit, and sequenced on a MiSeq instrument, using 2 × 250 bp chemistry (Illumina, San Diego, CA, USA). Illumina reads were mapped against a reference sequence (GenBank accession number HQ537483), using Bowtie2, and the consensus sequence was generated with ivar consensus (−q 20, −t 0.5, −m 10), available in Galaxy ARIES platform (https://aries.iss.it) [41]. Genetic lineage was assigned using Genome Detective version 1.2 (https://www.genomedetective.com/app/typingtool/wnv/) [42,43]. The complete coding sequence (CDS) of WNV genomes was investigated for amino acid substitutions described as molecular markers for virulence in mammalian and avian hosts reviewed elsewhere [44].

### 2.8. Phylogenetic Analysis

The phylogenetic tree was built with MEGA11, using the Neighbor-Joining method and Kimura 2-parameter method, 1000 bootstrap replicates [45]. Ten sequences displaying suboptimal lengths were excluded from the phylogenetic analysis to achieve a satisfactory tree resolution. Tree annotations were added with EvolView v3 [46].

Sequences obtained in this study were deposited in GenBank under the accession numbers PP524755, PP524756, PP533480-PP533488, PP533490-PP533530, and PQ124098-PQ124102.

This study was approved by the Ethics Committee of the Cantacuzino National Military Medical Institute for Research and Development (ethical approval codes CE332/2019 and CE208/2022).

## 3. Results

During the study period, 1604 mosquito pools comprising 57,391 specimens were screened for the presence of WNV. Excluding the 90 pools corresponding to the collections performed in 2022, which were analyzed by RAMP assay due to the personnel shortage encountered that year, all other samples were tested by Real-Time PCR, followed by Sanger sequencing. The collected specimens belonged mainly to *Cx. pipiens* s.l. species (92.1%), and to a lesser extent to the *Aedes albopictus* species (7.9%). WNV was detected in 139 pools containing *Cx. pipiens* s.l. specimens (132 pools by Real-Time PCR and seven pools by RAMP assay), and in eight *A. albopictus* pools (by Real-Time PCR) (Table 1 and Appendix A). Minimum infection rate calculated for *Cx. pipiens* s.l. (MIR—i.e., [number of positive pools/total specimens tested] × 1000) ranged between 0.77 in 2020 and 7.57 in 2017 (Table 1). The trend line of MIR values calculated for the study period correlated with the trend line of human infection cases reported during the same time interval in Romania, except for 2017 and 2021, when the high values of MIR were not reflected in the number of human cases (Figure 2). Except for in 2019, when the first mosquito samples testing positive for WNV were detected at the beginning of August, for the rest of the years of the study, WNV-positive mosquito samples were recorded beginning with the first days of July or at the end of June 2017 (Appendix A).

Of the 140 mosquito pools testing positive for WNV in the Real-Time PCR test, 52 yielded a NS5 partial sequence (Ct values between 13.82 and 30.43). For two samples collected in 2023 whole genome sequences are available (Table 1 and Appendix A). Sequences belonged to cluster B (*n* = 49) and to cluster A (*n* = 2) of sub-lineage 2a, but also to sub-lineage 2b (*n* = 1) (Figure 3 and Figure 4, Appendix A).

Of the 91 samples originating from serologically confirmed WNV human patients, 19 samples recovered from the 2017 outbreak and 16 samples from the 2018 outbreak were Real-Time PCR positive for WNV, respectively. Four samples yielded NS5 partial sequences (Ct values between 13.89 and 27.00), all belonging to cluster B of sub-lineage 2a (Figure 1, Figure 3 and Figure 4, Appendix A).

Regarding the bird samples, only the serum from the hooded crow specimen collected in Bucharest city during the epidemics of 2018 was Real-Time PCR-positive for WNV (Ct 25.35). Specifically, the sample generated a NS5 partial sequence belonging to an isolate of sub-lineage 2a, cluster B (Figure 1, Figure 3 and Figure 4, Appendix A).

The only sequence derived from a sub-lineage 2b WNV isolate (i.e., eastern European clade of lineage 2, also named Russian/Romanian clade, prototype strain Volgograd 2007) was obtained from *Cx. pipiens* mosquitoes collected in 2017 from a site located in the suburbs of the Bucharest city. The sequence is highly similar to the sequences obtained between 2011 and 2015 in southeastern Romania from mosquitoes, ticks, and human patients, with a mosquito sequence in 2014 from Italy, and with more recent sequences in 2023 from Krasnodar, Russia (Figure 4, Appendix A).

Furthermore, it is worth mentioning that the presence of WNV isolates belonging to cluster A of sub-lineage 2a was evidenced for the first time in Romania. Those two sequences derived from two pools of mosquitoes collected in 2020 in two different locations in Bucharest city, and are closely related to a 2021 human isolate from a Hungarian patient, but they also share similarities with bird, mosquito, and human isolates from Central Europe, Italy, Spain, and Serbia identified in the last two decades (Figure 4, Appendix A).

Most of the sequences obtained in this study belong to cluster B of sub-lineage 2a. However, for most of the years of the study, these sequences were split between two sub-clusters or, as in the case of 2021, between three sub-clusters. The first sub-cluster comprises isolates circulating in southeastern Romania (2015–2023, except 2020), Bulgaria (2018), and Greece (2019–2022). The second sub-cluster contains sequences from central Romania (2015), southeastern Romania (2018–2023), eastern Romania (2019), Hungary (2018), Serbia (2018), Greece (2018 and 2019), but also Poland (2022). The third sub-cluster contains sequences from southeastern Romania (2018 and 2021), eastern Romania (2018), sequences from a traveler in Romania returning to Spain (2018) and from a Belgian traveler returning from Hungary (2017), and sequences from mosquitoes and humans in Greece (2018) as well. The sequence from the human case in 2017 was found in the first sub-cluster, while the 2018 human sequences were found in the first two sub-clusters. The bird-derived sequence (*C. cornix*, Bucharest city, 2018) is placed in the first sub-cluster (Figure 4, Appendix A).

It is worth mentioning that different WNV strains were found at the same collection site, even in the same year. WNV-2b and WNV-2a isolates were found co-circulating in a site studied in 2017. Isolates belonging to different sub-clusters of cluster B (WNV-2a) were found co-circulating in two other distinct sites analyzed both in 2019 and 2021 (Figure 4, Appendix A).

Whole genome sequences were obtained for only two mosquito isolates from 2023, which were considered representative of the first two sub-clusters described above, after previous sequencing and analysis of the NS5 fragment. The CDS of the two genomes were 99.71% identical at nucleotide level and differed by only three amino acid residues. In both genomes, two substitutions previously described as virulence determinants were found at positions 159 in envelope glycoprotein and at position 249 in NS3 protein, respectively (Appendix A).

Interestingly, WNV was detected in three *Cx. pipiens* male pools from 2018. However, WNV sequences could not be obtained from these pools (Ct values > 33) (Appendix A).

WNV was detected in eight *A. albopictus* female mosquito pools, but due to the low quantity of virus in the samples (Ct values > 31), no sequence was obtained (Appendix A).

## 4. Discussion

Except in 2002, when no cases were reported by the national authorities, WNV infection cases in the Romanian human population were serologically documented yearly for almost three decades.

The present study aims to fill the information gap regarding the WNV strains circulating in Romania between 2017 and 2023, providing chiefly viral sequences obtained from mosquito samples, complemented by human and bird viral sequences.

The mosquito specimens collected during these seven years across Bucharest city indicated that *Cx. pipiens* s.l. and *A. albopictus* mosquitoes were testing positive for WNV by Real-Time PCR assay. Moreover, MIR was calculated for *Cx. pipiens* s.l. and correlated with the number of human disease cases recorded in Romania yearly for the study period, except for 2017 and 2021. However, the low number of human cases in 2021 might have been linked with the sanitary crisis produced by the COVID-19 pandemic, which led to the underdiagnosis of human cases of WNV, or it might also have been a consequence of the significant reduction of outdoor activities and traveling determined by the prevention measures imposed to limit SARS-CoV-2 transmission, as suggested elsewhere [47]. European Union (EU) and European Economic Area (EEA) Member States and EU-neighboring countries reported a significantly lower number of cases in 2021 compared with the previous five years [48]. The highest value of MIR was calculated in 2017, the year before a significant outbreak in Romania. This is in accordance with a previous observation, indicating that after first introduction, a 1- to 2-year period of silent enzootic transmission of WNV precedes the spread to humans [17]. In contrast, the lowest MIR value (0.77) was recorded for 2020, indicating a reduced circulation of WNV in mosquito populations, which might have been shaped by environmental factors such as temperature and precipitation, availability of breeding sites, land use and land cover, etc. [35].

*Cx. pipiens* s.l. species was previously documented as an urban vector of WNV in Romania [9,22], with transmission cycles shaped by its two biotypes, namely *pipiens* and *molestus*, and their hybrids. Due to its feeding preferences towards avian hosts, the *pipiens* biotype acts as an enzootic/epizootic vector, while the *molestus* biotype, which feeds similarly on both birds and mammals, represents the bridge vector [49]. However, entomological studies carried out in the Danube Delta found other mosquito species infected with different lineages of WNV (e.g., *Culex modestus* and *Uranotaenia unguiculata*), including the Volgograd 2007-like epidemic strain, which may also have played a part in the sylvatic transmission cycles in Romania [22]. The Danube Delta is an important stopover for migratory birds [50], and plays a significant role in WNV transmission cycles, as demonstrated by several other studies which detected WNV in mosquitoes [51,52], birds [53,54], semi-feral horses [55], and ticks [20]. It was shown that WNV circulation in mosquito populations from the Danube Delta is intensified by weather conditions such as positive temperature anomalies in spring and summer and rainfall decrease [51].

During the study period, only lineage 2 WNV was identified in mosquito samples collected in the Bucharest city area, a finding also observed in the human sera collected in southern and central Romania during the 2017 and 2018 outbreaks.

A single sequence belonging to sub-lineage 2b (the Eastern European clade also known as Russian/Romanian clade of lineage 2) was obtained in 2017 from *Cx. pipiens* mosquitoes collected in a single site located in Bucharest city. The first isolate of this sub-lineage was detected in 2007 in the brain of a human patient from an outbreak in the Volgograd region, Russia. This strain was subsequently recorded during major epidemics in 2010 in the Volgograd and Rostov regions, Russia [56], and Romania [21]. Between 2011 and 2016, similar isolates were found in human patients from the southeastern counties of Romania, but also in mosquitoes collected in Bucharest city and the Danube Delta [20,22,52]. Only two studies have identified the Volgograd 2007 strain circulating in countries other than Russia and Romania. In 2014, the strain was identified in *Cx. pipiens* mosquitoes collected from two sites 50 km apart in northeastern Italy, but whether the strain was introduced by birds migrating from Africa or through a shorter migration from Eastern Europe remained unknown [57]. More recently, during the unprecedented WNV epidemics in 2018, a Volgograd 2007-like isolate was detected in one human case in the northeastern region of Greece, at the land cross-border with Turkey and Bulgaria, suggesting a novel virus introduction in Greece, a country where sub-lineage 2a has been dominant since the 2010 outbreak [58]. Currently, the most recent Volgograd 2007-like sequences are from the 2023 transmission season in Krasnodar, Russia, as shown by the publicly available sequences. A study on the ecology of WNV in the Danube Delta, Romania [52], concluded that the introduction of this sub-lineage 2b strain in the region was a single event produced during the 1990s, the strain being a descendant of an ancestor that probably emerged in South Africa around 1910, which is consistent with a previous study [20], and its genetic diversity was shaped by in situ evolution.

The other sequences described in our seven-year study belong to sub-lineage 2a, which comprises more than 70% of the WNV sequences obtained in Europe and shared in the public domain [16]. All the human sequences, the bird sequence, and the vast majority of the mosquito sequences described in the present study are part of cluster B of sub-lineage 2a. In our NS5-based phylogeny, cluster B splits into three sub-clusters. A previous Romanian study [23] indicated the existence of two sub-clusters containing sequences from southeastern Romania and central Romania, respectively. This geographic segregation is no longer seen in the present analysis, since the sequences originating from southeastern and central Romania and those from eastern Romania, generated in this study and in others [23,59,60], are mixed in two out of the three of the identified sub-clusters.

A sequence obtained from mosquitoes collected in 2018 in a county neighboring Bulgaria (our group, unpublished data), a sequence from mosquitoes collected eastern Romania in 2018 [59], and a sequence from mosquitoes collected in Bucharest city in 2021 (this study) are part of a third sub-cluster, together with a sequence obtained from a traveler returning from Romania in 2018 [60], as well as with sequences from human and mosquitoes from Greece (2018), and with a sequence from a Belgian traveler returning from Hungary (2017). Similar clustering of the Greek and Hungarian sequences was demonstrated in a previous study [17], showing that in 2018, three novel, independent introductions from Hungary and Bulgaria were responsible for the 2018 re-emergence of WNV in northern Greece during the unprecedented outbreak that year [61]. This particular sub-cluster which contains the Romanian sequences also originated from a novel introduction from Hungary. Furthermore, the second sub-cluster described in our analysis contains sequences from isolates circulating in 2018 in northern Greece, and which were considered the result of another introduction of WNV into Greece from Hungary in 2018 [17]. However, this sub-cluster also contains a sequence from a human isolate from central Romania [23], thus we concluded that this sub-cluster of isolates was already present in Romania prior to 2018, but we cannot speculate whether the virus migrated from central Romania to southeastern parts of the country, or whether we are facing a distinct introduction from another country, like in the case of Greece. The diversity of cluster B isolates circulating between 2015 and 2023 in Romania are, as was shown previously [16], the effect of intense transmissions (jumps) between countries and regions. The latter study showed transmission events between Romania and Greece taking place both ways.

Only two sequences, obtained in 2020 from mosquitoes collected in Bucharest, belonged to cluster A of sub-lineage 2a. It was shown that this cluster has a lower dispersal velocity than cluster B, which tends to jump almost twice as frequently among countries [16].

The analysis of WNV genomes identified in mosquitoes collected in Bucharest city in 2023 showed the presence of the amino acid substitution H249P in NS3 protein and of I159T in E glycoprotein, mutations previously found in the isolates circulating in Romania in the period of 2015–2016 [23]. H249P substitution was first detected in the Greek Nea Santa-2010 strain, and was presumptively considered a virulence marker [14]. Previous studies associated the presence of proline at position 249 in NS3 protein encoded by lineage 1 NY99 strain with high viremia and mortality rate in birds [62,63]. However, another study evaluating the pathogenicity of WNV lineage 2 strains found that NS3-249P mutation is neither sufficient nor necessary for conferring virulence, nor is it necessary to confer pathogenicity to any given lineage 2 WNV strain in birds [64]. It was shown that NS3-249 site in lineage 2 WNV strains circulating in Europe between 2004–2018 was not under positive selection pressure, and H249P represents a signature of the Balkan subgroup of strains [17].

Interestingly, in the current study, WNV was detected in three pools of *Cx. pipiens* s.l. mosquito males collected in 2018 in Bucharest city, indicating virus vertical transmission. Unfortunately, the low viral load in the mosquito samples did not allow for obtaining viral sequences. This is the second round of field evidence indicating WNV vertical transmission in the same area [22]. A study showed that WNV from naturally infected *Cx. pipiens* female was vertically transmitted to a female offspring, which in laboratory conditions showed vector competency, thus suggesting that WNV survives winter in unfed, vertically infected mosquitoes [65].

The second species found positive for WNV by Real-Time PCR in our study was *A. albopictus*, a species rated as the most invasive mosquito species, and one of the top invasive organisms in the world. This mosquito is involved in local transmission of chikungunya and dengue viruses in Europe [66]. Introduction of *A. albopictus* in Romania was documented in 2012 in Bucharest city, with populations similar to the ones in temperate and subtropical regions [67]. Subsequent entomological surveys indicated the presence of *A. albopictus* in eight counties of Romania (2020) [68], and a northward dispersal of the species [69]. The vector competence of *A. albopictus* to WNV was shown experimentally [70,71], and several studies employing PCR or cell culture assays for WNV detection performed on mosquito population from US [72,73,74], Turkey [75], and Greece [76] also demonstrated natural infection.

Our seven-year study provides a glimpse into the genetic diversity of the WNV isolates circulating in mosquito populations in the Bucharest city area and human patients during the outbreaks of 2017 and 2018 in Romania. The similarity of isolates belonging to cluster B of sub-lineage 2a circulating in recent years with those first detected in 2015–2016 in the area indicates continuous circulation followed by in situ evolution. However, the existence of different sub-clusters of viral isolates comprising isolates with mixed geographic origins sustains the hypothesis of multiple introductions, occurring especially during the high epidemic years, such as 2018, from neighboring countries. It was shown that WNV spreads to areas with a high degree of urbanization, as is the case for our study area, where less balanced mosquito communities and higher abundance of *Cx. pipiens* are recorded [16]. Phylogeography analysis indicates that WNV circulates in reservoir and mosquito vector species years before animal and human outbreaks, and suggests that even in the absence of human cases, surveillance systems should be maintained in high-risk areas [18]. Active genomic surveillance of WNV in mosquito populations serves as such a surveillance system. Partial genomic sequences, such as those employed in our molecular study, can only provide limited information, and, as recommended [3], genomic surveillance of WNV should be strengthened at the continental level in Europe, especially with the recent unprecedented level of northern expansion, using the infrastructure developed during COVID-19 pandemic, transitioning from partial to whole genome sequencing. Complete viral genome sequences are the premises for high-resolution phylogeography and phylodynamics studies, and can be used to identify lineage-specific mutations that impact transmission potential and virulence [3,16,77]. Deep-sequencing technologies provide more insights for WNV evolution and intra- and inter-host adaptation than Sanger consensus sequences, which poorly depict the mutational landscapes and do not accurately reflect diversity hotspots, substitution bias, or host-specific selective pressures [78].

## Figures and Tables

**Figure 1 microorganisms-12-02080-f001:**
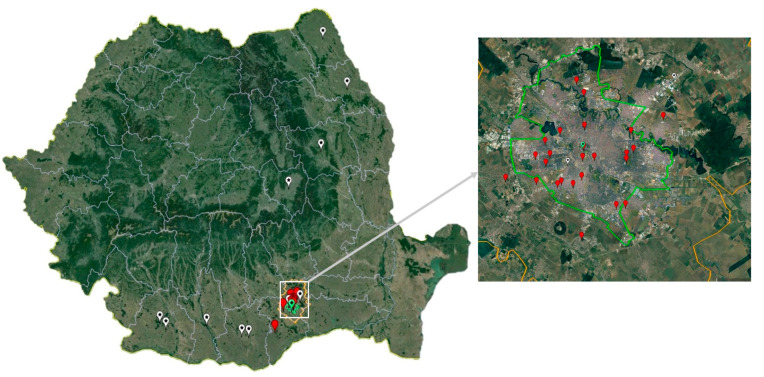
Mosquito and bird sampling sites (Bucharest city area, 2017–2023) and West Nile virus human infection cases confirmed by Real-Time PCR (2017 and 2018). White pin: human infection case; red pin: mosquito sampling site; green pin: *Corvus cornix* sampling site.

**Figure 2 microorganisms-12-02080-f002:**
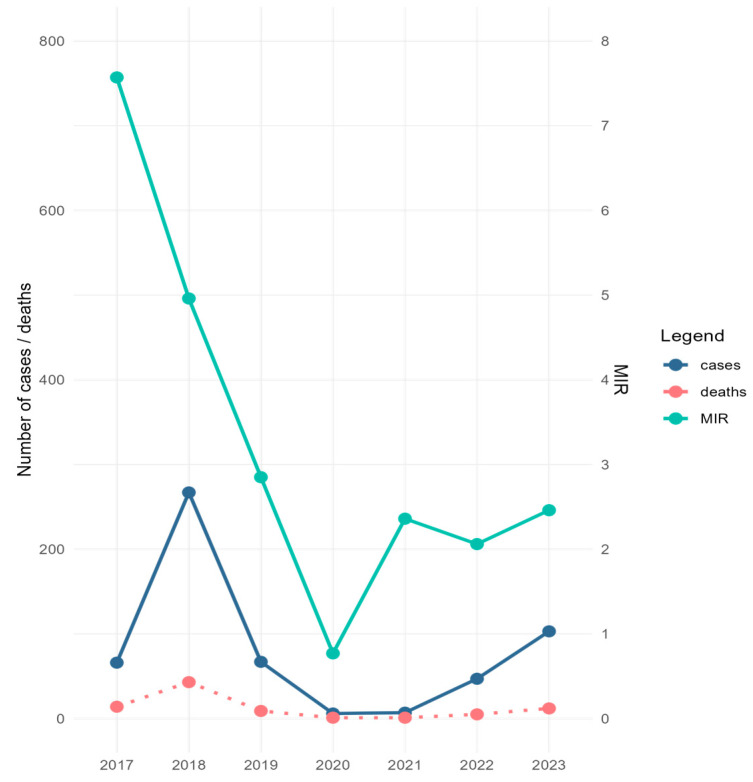
Minimum infection rate (MIR) of West Nile virus (WNV) calculated for *C. pipiens* s.l. mosquitoes collected in Bucharest city area, 2017–2023, and number of WNV human disease cases/deaths for the same period in Romania. MIR = (number of positive pools/total specimens tested) × 1000.

**Figure 3 microorganisms-12-02080-f003:**
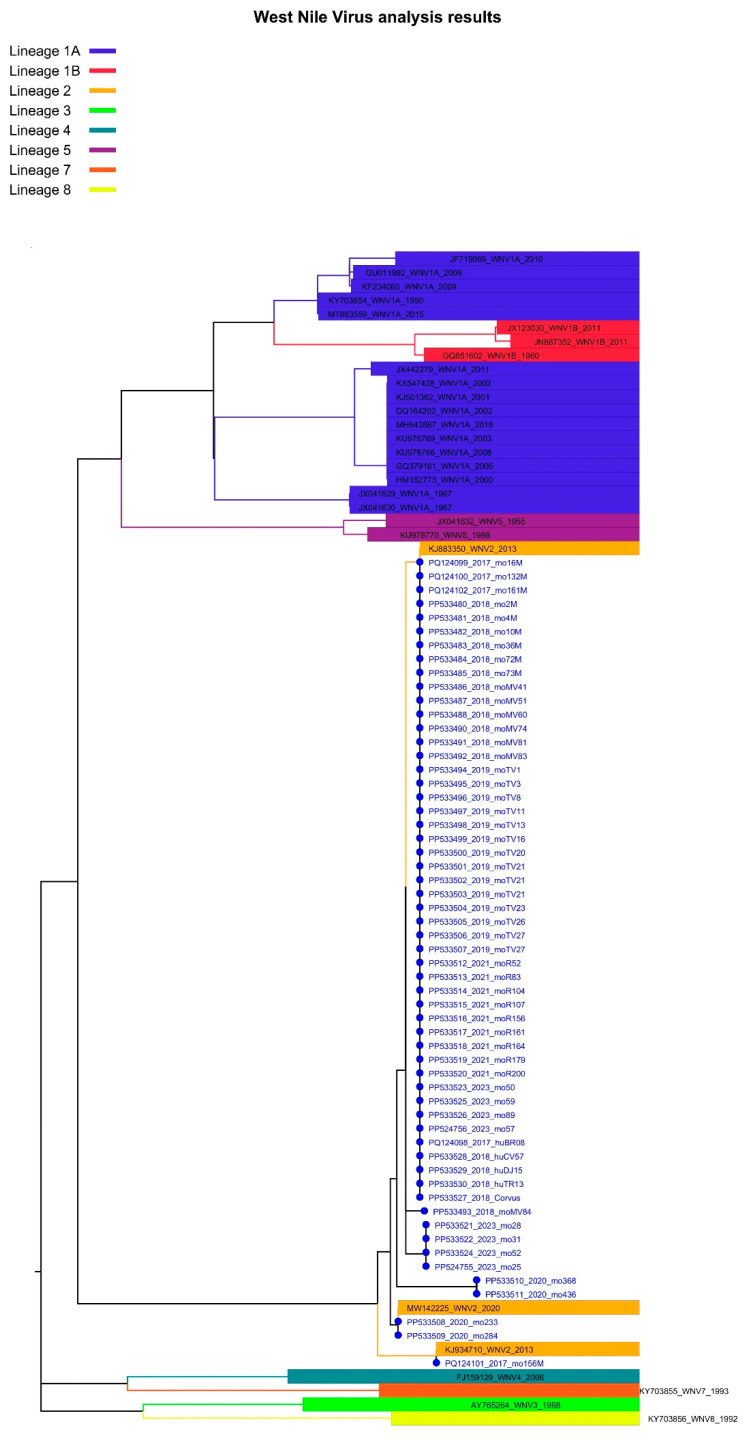
Lineage assignment for West Nile virus isolates identified in Romania, 2017–2023, using Genome Detective version 1.2 (https://www.genomedetective.com/app/typingtool/wnv/).

**Figure 4 microorganisms-12-02080-f004:**
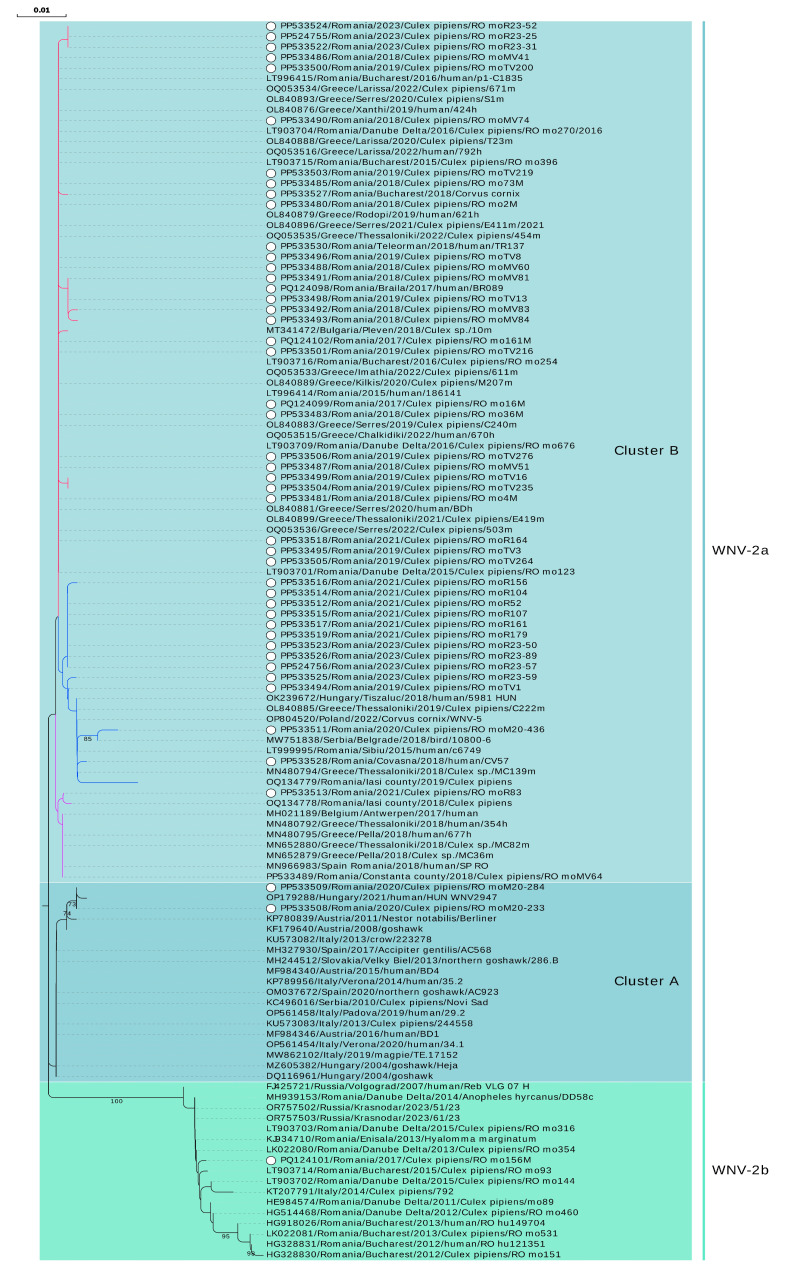
Phylogenetic analysis of West Nile virus isolates identified in Romania, 2017–2023. The evolutionary history was inferred from a NS5 fragment (nucleotides 9476–10,000 in Nea-Santa-2010 genome, GenBank acc. no. HQ537483) using the Neighbor-Joining method and Kimura 2-parameter method. Rate variation among sites was modeled with a gamma distribution (shape parameter = 0.18). The percentage of replicate trees in which the associated taxa clustered together in the bootstrap test (1000 replicates) are shown next to the branches (values < 70% are not shown). White dot: sequence obtained in this study. Branches of sub-clusters belonging to cluster B of sub-lineage 2a are marked with different colors. Nomenclature for lineages and clusters used is described in [16].

**Table 1 microorganisms-12-02080-t001:** Summary of the mosquitoes collected and tested for West Nile virus (WNV), Bucharest area, Romania, 2017–2023.

Year	*C. pipiens* s.l.		*A. albopictus*	
	Collection *	MIR **	Collection	MIR
2017	4884/170/37/4	7.57	1957/64/8/0	4.08
2018	5436/156/27/13	4.96	1699/60/0/0	0.00
2019	8060/223/23/14	2.85	59/2/0/0	0.00
2020	18,156/486/14/4	0.77	231/6/0/0	0.00
2021	8878/210/21/9	2.36	433/9/0/0	0.00
2022	3384/90/7/0	2.06	-	-
2023	4051/124/10/8	2.46	163/4/0/0	0.00
Total	52,849/1459/139/52		4542/145/8/0	

* Number of mosquitoes/number of mosquito pools/number of mosquito pools positive for WNV/number of WNV sequences; ** Minimum infection rate: (number of WNV positive pools/total specimens tested) × 1000.

## Data Availability

The original contributions presented in the study are included in the article/Appendix A, further inquiries can be directed to the corresponding author.

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
