# Peer review of "Continuous and Dynamic Circulation of West Nile Virus in Mosquito Populations in Bucharest Area, Romania, 2017–2023"

_microorganisms, 2024, doi:10.3390/microorganisms12102080_

Round 1

Reviewer 1 Report

Comments and Suggestions for Authors

This study examines West Nile Virus (WNV) strains from 2017 to 2023, focusing primarily on mosquito vectors, including Culex pipiens and Aedes albopictus, along with viral sequences detected in humans and birds, which adds substantial depth to the research. The distribution of mosquito species tested is clearly presented, with Culex pipiens s.l. comprising 92.1% and Aedes albopictus 7.9%, reflecting the diversity of vector populations involved in the study. Additionally, the study addresses key gaps by highlighting recent trends in WNV circulation and transmission in Romania.

I made only a few suggestions:

In the introduction, It would be valuable to emphasize the importance of tracking WNV circulation, particularly regarding its implications for public health interventions. This could include strategies for disease control, vaccination efforts, and vector management, which are becoming increasingly critical in response to the climate-driven expansion of WNV transmission.

The inclusion of Figure S1 directly within the main body of the paper would enhance the clarity and accessibility of the data.

While the results are comprehensive, providing greater clarity on the criteria used for selecting pools for Real-Time PCR versus the RAMP assay would enhance the transparency of the methodology.

Finally, the authors suggest expanding from partial to whole-genome sequencing, which would provide more comprehensive data on viral evolution and transmission patterns—a point with which I fully agree. I believe this discussion could be further expanded by including other areas for future study, such as exploring genetic diversity across different regions or investigating host-pathogen interactions in greater detail.

Author Response

We take the opportunity to thank to reviewer #1 for the time dedicated to evaluate our work. The comments and suggestions provided improved the quality of our MS. Please find below our point-to-point answers.

Comments and Suggestions for Authors

This study examines West Nile Virus (WNV) strains from 2017 to 2023, focusing primarily on mosquito vectors, including Culex pipiens and Aedes albopictus, along with viral sequences detected in humans and birds, which adds substantial depth to the research. The distribution of mosquito species tested is clearly presented, with Culex pipiens s.l. comprising 92.1% and Aedes albopictus 7.9%, reflecting the diversity of vector populations involved in the study. Additionally, the study addresses key gaps by highlighting recent trends in WNV circulation and transmission in Romania.

I made only a few suggestions:

In the introduction, It would be valuable to emphasize the importance of tracking WNV circulation, particularly regarding its implications for public health interventions. This could include strategies for disease control, vaccination efforts, and vector management, which are becoming increasingly critical in response to the climate-driven expansion of WNV transmission.

R: Importance of tracking WNV circulation and its implications for public health interventions was discussed in the revised MS, Introduction section, lines 94-99.

The inclusion of Figure S1 directly within the main body of the paper would enhance the clarity and accessibility of the data.

R: Figure S1 is now included in revised MS as Figure 3, page 7.

While the results are comprehensive, providing greater clarity on the criteria used for selecting pools for Real-Time PCR versus the RAMP assay would enhance the transparency of the methodology.

R: In 2022, due to the personnel shortage, the WNV mosquito surveillance was done using only the RAMP assay. This statement in now include in the revised MS, the Results section, lines 185-186.

Finally, the authors suggest expanding from partial to whole-genome sequencing, which would provide more comprehensive data on viral evolution and transmission patterns—a point with which I fully agree. I believe this discussion could be further expanded by including other areas for future study, such as exploring genetic diversity across different regions or investigating host-pathogen interactions in greater detail.

R: The Discussion section was expanded as suggested, lines 464-470, revised MS.

Reviewer 2 Report

Comments and Suggestions for Authors

The paper by Dinu and colleagues provides inoformation on the WNV sequences identified during the period 2017/23 in the Bucharest region in Romania.
The brief introduction provides an historycal overview of the spread of WNV in Romania up to the date of the study's observation
The methods section is also very brief, but sufficinelty explanatory (please see the minor points section)
The results are sufficiently well presented, except for a few minor (but relevant) points. A limitation of the study, as the results show, is the absence of human samples from the seasons followinf 2018, but
it looks like this doesn't affect the general merit of the paper.
The discussion is really too long. Part of it deals with background information that is overlapping with the introduction, part is repetitive with the results section. In any case, the long historycal overview of
the sublineages introduction and circulation in Europe is too long, and more aapropriate for a review than for the discussion of a research paper (just to provide a measure, the discussion is one third of the whole paper).

Some minor points:
- line 79 and following: it would be useful to remind that in the rest of Europe, along with lineage 2, recent seasons have shown the reintroduction of lineage 1
- line 103: it is not quite clear why no human samples were included from the following seasons
- line 119: I suggest to benefit the readers by adding at least a very general description of the test
- line 124: maybe the paragraph should be splitted between sanger and WGS, or at least there should be better distinction on which analysis have been done on which type of sequences
- line 152: ethical approval ID should be provided
- line 167: actually, the most striking exception to the correlation is for year 2017, and this should be noted and, if possible, explained. Furthemore, it is not clear if the human cases reported
are from the same region as the mosquito pool, or from the whole country (the first option being the most preferable one). In any case, please state it in the text
- line 179: Are these mosquitos from the whole country or from the region depicted in Fig.1? Please state it in the text
- line 280: mainly, except for 2017
- line 296: please list some of these other mosquito species

Author Response

We take the opportunity to thank to reviewer #2 for the time dedicated to evaluate our work. The comments and suggestions provided improved the quality of our MS. Please find below our point-to-point answers.

Comments and Suggestions for Authors

The paper by Dinu and colleagues provides information on the WNV sequences identified during the period 2017/23 in the Bucharest region in Romania.

The brief introduction provides an historical overview of the spread of WNV in Romania up to the date of the study's observation

The methods section is also very brief, but sufficiently explanatory (please see the minor points section)

The results are sufficiently well presented, except for a few minor (but relevant) points. A limitation of the study, as the results show, is the absence of human samples from the seasons following 2018, but

it looks like this doesn't affect the general merit of the paper.

The discussion is really too long. Part of it deals with background information that is overlapping with the introduction, part is repetitive with the results section. In any case, the long historical overview of

the sublineages introduction and circulation in Europe is too long, and more appropriate for a review than for the discussion of a research paper (just to provide a measure, the discussion is one third of the whole paper).

R: Thank you for the appreciation! We took your advice and shortened the Discussion section. Redundant background information overlapping with Introduction was completely removed (lines 293-302, 346-347, 367-369, 350-351, and 411-412) from the revised MS. The historical overview of the sub lineages was moved from Discussion section (lines 368-378) to Introduction section (lines 64-68), revised MS.

Some minor points:

- line 79 and following: it would be useful to remind that in the rest of Europe, along with lineage 2, recent seasons have shown the reintroduction of lineage 1

R: Information added in Introduction section, revised MS (lines 89-90).

- line 103: it is not quite clear why no human samples were included from the following seasons

R: this limitation has been addressed in the revised MS, Materials and Methods section (lines 121-122).

- line 119: I suggest to benefit the readers by adding at least a very general description of the test

R: the description of the RAMP test was added, Materials and Methods section of the revised MS (lines 136-141).

- line 124: maybe the paragraph should be splitted between sanger and WGS, or at least there should be better distinction on which analysis have been done on which type of sequences

R: The paragraph was split into “2.6 Sanger sequencing”, “2.7 Whole genome sequencing”, and “2.8 Phylogenetic analysis”, revised MS (lines 142-175).

- line 152: ethical approval ID should be provided

R: Ethical approval codes were added at end of Materials and Methods section, revised MS (lines 180-181).

- line 167: actually, the most striking exception to the correlation is for year 2017, and this should be noted and, if possible, explained. Furthermore, it is not clear if the human cases reported are from the same region as the mosquito pool, or from the whole country (the first option being the most preferable one). In any case, please state it in the text

R: This was added in the Results (line 195) and Discussion (line 310) sections of the revised MS. The 2017 exception was already discussed in the original MS, Discussion section (now lines 317-320 in the revised MS). The cases used for MIR calculation are from the whole country (piece of information added in the revised MS, Figure 2 caption, line 212 and already present in the original MS, Discussion section, now line 309 in the revised MS).

- line 179: Are these mosquitos from the whole country or from the region depicted in Fig.1? Please state it in the text

R: the mosquitoes are from Bucharest area, the requested information was added at lines 200-201, revised MS.

- line 280: mainly, except for 2017

R: information added, line 310, revised MS.

- line 296: please list some of these other mosquito species

R: information added, Discussion section line 330, revised MS.

Reviewer 3 Report

Comments and Suggestions for Authors

The manuscript: "Continuous and dynamic circulation of West Nile virus in mosquito populations in Bucharest area, Romania, 2017-2023" is a valid addition to all the other publications by this group of authors. Introduction and methods need no improvement as these parts are written very thoroughly. Some concerns rise with the study population; while mosquito samples originate from all the years this research covers, human and bird samples are only a few and belonging to a same time period so it is expected that the isolated strains are the same. The results are overall adequately presented, but it would be interesting for the authors to elaborate a bit more on why is MIR so low in 2020. Discussion contains enough studies with comparable results and conclusions are supported by the results. 

Comments on the Quality of English Language

The English language requires minor to moderate interventions which can be resolved within the editing process.

Author Response

We take the opportunity to thank to reviewer #3 for the time dedicated to evaluate our work. The comments and suggestions provided improved the quality of our MS. Please find below our point-to-point answers.

Comments and Suggestions for Authors

The manuscript: "Continuous and dynamic circulation of West Nile virus in mosquito populations in Bucharest area, Romania, 2017-2023" is a valid addition to all the other publications by this group of authors. Introduction and methods need no improvement as these parts are written very thoroughly. Some concerns rise with the study population; while mosquito samples originate from all the years this research covers, human and bird samples are only a few and belonging to a same time period so it is expected that the isolated strains are the same. The results are overall adequately presented, but it would be interesting for the authors to elaborate a bit more on why is MIR so low in 2020. Discussion contains enough studies with comparable results and conclusions are supported by the results. 

R: The low MIR value for 2017 was commented in the Discussion section of the revised MS, lines 320-323. Unfortunately, we do not have data on the environmental factors that could have affected the circulation of WNV in the mosquito population that year.

Reviewer 4 Report

Comments and Suggestions for Authors

This manuscript is valuable for the vector and epidemic control of West Nile virus, but it has some shortcomings:

1.Figure 3 needs to be reworked so that it is focused and beautiful.

2.Additional haploid phylogenetic analysis is recommended.

3.The result section needs to be refined and rewritten.

4.The discussion section needs to refine the sulk.

Comments on the Quality of English Language

/

Author Response

We take the opportunity to thank to reviewer #4 for the time dedicated to evaluate our work. The comments and suggestions provided improved the quality of our MS. Please find below our point-to-point answers.

Comments and Suggestions for Authors

This manuscript is valuable for the vector and epidemic control of West Nile virus but it has some shortcomings:

1.Figure 3 needs to be reworked so that it is focused and beautiful.

R: Figure 3 was provided at the requested resolution.

2.Additional haploid phylogenetic analysis is recommended.

R: We believe that viral haplotype analysis is not suitable for our study since the overwhelming majority of the sequences obtained are Sanger sequences and not obtained by deep-sequencing technologies.

3.The result section needs to be refined and rewritten.

R: The results section was refined and information requested by the other reviewers was added.

4.The discussion section needs to refine the sulk.

The Discussion section was modified. Redundant background information overlapping with Introduction was completely removed (lines 293-302, 346-347, 367-369, 350-351, and 411-412) from the revised MS. The historical overview of the sub lineages was moved from Discussion section (lines 368-378) to Introduction section (lines 64-68), in the revised MS. Additional comments on the importance of complete genome sequences and deep-sequencing for WNV study were added, lines 464-470, in the revised manuscript.

Round 2

Reviewer 4 Report

Comments and Suggestions for Authors

The author has made a large revision, solved some problems, and proposed to be published.

Comments on the Quality of English Language

/